# Influence of Implant Scanbody Wear on the Accuracy of Digital Impression for Complete-Arch: A Randomized In Vitro Trial

**DOI:** 10.3390/ma15030927

**Published:** 2022-01-25

**Authors:** Lorenzo Arcuri, Fabrizio Lio, Veronica Campana, Vincenzo Mazzetti, Francesca Romana Federici, Alessandra Nardi, Massimo Galli

**Affiliations:** 1Materials for Health, Environment and Energy, University of Rome Tor Vergata, 00133 Rome, Italy; lorenzo.arcuri@uniroma2.it (L.A.); vincenzo.mazzetti1992@gmail.com (V.M.); 2Independent Researcher, 20121 Milan, Italy; dr.campanaveronica@gmail.com; 3Innovative Technologies in Skeletal, Skin and Oro-Cranio-Facial Diseases, Sapienza University of Rome, 00185 Rome, Italy; fr.federici@yahoo.it; 4Department of Mathematics, University of Rome Tor Vergata, 00133 Rome, Italy; alenardi@axp.mat.uniroma2.it; 5Department of Dental and Maxillofacial Sciences, Sapienza University of Rome, 00185 Rome, Italy; massimo.galli@uniroma1.it

**Keywords:** digital impression, intraoral optical scanning, complete arch, scanbody, accuracy, wear

## Abstract

The aim of this study was to evaluate the influence of implant scanbody (ISB) wear on the accuracy of digital impression for complete-arch. A polymethylmethacrylate (PMMA) edentulous mandibular model with four internal hexagonal interlocking conical connections was scanned with an extraoral optical scanner to achieve a reference file. Four cylindrical polyetheretherketone (PEEK) ISBs were scanned 30 times with IOS, and the test files were aligned to the reference file with a best-fit algorithm. For each analog linear (ΔX, ΔY and ΔZ-axis) and angular deviations (ΔANGLE) were assessed. Euclidean distance (ΔEUC) was calculated from the linear deviation, reporting a mean of 82 µm (SD 61) ranging from 8 to 347 µm. ΔANGLE error mean was 0.33° (SD 0.20), ranging from 0.02 to 0.92°. From a multivariate analysis, when ΔEUC was considered as a response variable, a significant influence of ISB wear by scan number in interaction to position for implant 3.6 was identified (*p* < 0.0001); when ΔANGLE was considered as a response variable, a significant effect of position 3.6 was recorded ((*p* < 0.0001). The obtained results showed that the ISB wear negatively influenced the accuracy of IOS, suggesting that ISB base wear could be detrimental for the seating of ISBs on angulated implants.

## 1. Introduction

Dental impressions certainly represent a crucial step in implant dentistry [1]. Inaccurate record and transfer of the implant position can lead to an ill-fitting prosthesis, which may result in both biological and mechanical complications [2]. Passive fit, defined as a clinical condition in which the prosthesis is not responsible for static loads in the prosthetic system or in the surrounding bone tissue, directly depends on the accuracy of the impression technique [3,4,5]. As defined by ISO-5725-1:1994, the accuracy of intraoral digital optical scanning (IOS) consists of trueness and precision. Trueness means the deviation from the true dimensions of the object, while precision describes how much different scans of the same object differ from each other [6,7]. Currently, IOS is potentially considered an alternative to the conventional implant impression to improve patient comfort and speed up workflow [8,9], advantageous features that are also applicable in the field of orthodontics and traditional prosthesis [10,11]. Furthermore, it was demonstrated that digital impressions are sufficiently accurate in tiny spaces; however, in more wide spaces such as those typically associated with edentulous patients, the results are less impressive in terms of both accuracy and speed [6]. The lack of stable reference points, teeth or keratinized mucosa; the minor amount of attached gingiva; and unfavorable conditions such as the presence of saliva, tongue and cheek movements all represent sources of inaccuracy [12,13]. Implant scanbody (ISB) are supplied to determine intraorally a positioning and orientation of an osseointegrated implant [3]. An ISB is commonly made up of three distinct components: the scan region, which is the upper one and responsible for digitally registering the three-dimensional orientation and angulation of an implant [14,15]; the body, which joins the scan region to the base; the base, which is the apical one and liable for creating the mating surface between implant and ISB. Generally, the scan region is made by the same material of the body, such as polyetheretherketone (PEEK), titanium alloy, aluminum alloy, and various resins. It is also characterized by one or multiple scan areas, but most commonly by a single flat side in order to index the ISB and improve the recognition by the CAD software [14,16]. Instead, the base can be made, or not, by the same material as the body [14]. A mismatch in materials between the base and the implant may influence displacement of the ISB when tightened into place [17]. The wear of this component through repeated use and sterilization may also cause changes in positioning over time, which could be problematic as the overall fit of any ISB is a decisive factor for a high-precision transfer of the implant position and inclination [18,19]. To our knowledge, the influence of wear resulting from ISB’s repositioning on the implant over the IOS impression accuracy was never investigated, and no reference data are currently available in the literature. The aim of this study was to evaluate the influence of ISB’s wear on the accuracy of the IOS impression.

## 2. Materials and Methods

### 2.1. Master Model

A polymethylmethacrylate (PMMA) edentulous mandibular model with 4 internal hexagonal interlocking conical connection implant analogs (NobelParallel RP 4.3, NobelBiocare, Kloten, Switzerland) was milled. The analogs were positioned at the sites of the right second molar, right canine, left central incisor and left first molar was produced, with the respective angulations and depths: 30° mesial angulation and 6.5 mm depth (4.7); 0° vestibular angulation and 7.35 mm depth (4.3); 0° and 0 mm depth (3.1); 17° distal angulation and 8.85 mm depth (3.6) (Figure 1). Bone resorption was simulated at the sites of analog 4.3 and 3.6, resulting in alveolar ridge drops. A distance of 22.8, 19.4 and 31.1 mm was made, respectively, between the implants 4.7–4.3, 4.3–3.1 and 3.1–3.6. A removable soft tissue was 3D printed (NextDent 5100, 3DSystems, Rock Hill, SC, USA) with a dedicated material (Gingiva Mask, NextDent, Soesterberg, Netherlands) to achieve accurate measurements with the reference scanner of the head of the analogs (Figure 1). Lastly, four polyetheretherketone (PEEK) ISBs cylinder shape with an axial incision and diameter and height, respectively, of 4.1 and 9 mm, were industrially produced with a ±0.01 mm tolerance (LaStruttura spa, Varese, Italy) by milling a PEEK dental disk (Invibio- Biomaterial solutions, Victrex, Hillhouse International, Thornton Cleveleys, Lancashire, England, UK) (Figure 2).

### 2.2. Reference Scan

An industrial 3D scanner (ATOS Compact Scan 5M, GOM GmbH, Braunschweig, Germany), based on structured blue light optical tech and previously calibrated, was used to obtain a reference standard tessellation language (STL) file by scanning the master model without the removable gingival tissue. A parametric measurement software (Gom Inspect Professional, GOM GmbH, Braunschweig, Germany) processed the model STL file. This elaboration allowed for combining geometrical coordinates of the analogs’ emerging platforms with analogs’ native mathematical files.

### 2.3. Intraoral Scanner and Scan Procedure

The cabled pen grip IOS investigated was a powder-free device based on confocal microscopy laser technology (Trios3, 3Shape A/S, Copenhagen, Denmark) with the software version 1.6.10.1. Thirty complete-arch scans were performed by a single operator, who underwent theoretical–practical training by scanning the test model ten times as recommended by the producer (3Shape A/S, Copenhagen, Denmark). The ISBs were screwed and unscrewed between the individual thirty scan sessions. The torque of 10 Ncm of ISBs onto the prosthetic connection was standardized for each screwing cycle and was controlled by a second operator means a dynamometer. Moreover, the correct coupling between ISBs bases and analog heads was visually checked through magnifying loupes (Eyezoom 5X, Orascoptic, Middleton, WI, USA). Every scan started from the analog in position 4.7, while the 3.6 was the last one to be scanned. Before the scan, 3D and color calibrations were performed as suggested by the producer. The scan strategy recommended by the producer was accomplished as follows: to prevent image splitting in the anterior area, a wave movement was made, and a 45° pen inclination indicatively was used. Firstly, ISB occlusal–palatal surfaces were scanned, followed by the buccal aspect and any other missing areas. Thirty test STL files were obtained by scan procedure (Figure 3).

### 2.4. Data Processing and Accuracy Assessment

The 30 test STL files were aligned to the reference scan through dedicated software (Geomagic Studio 12, 3DSystems, Rock Hill, SC, USA) to a 0.01 mm tolerance. After the superimposition, two alignment optimizations were performed (Figure 4). Therefore the linear geometries of the analogs were reconstructed, and subsequently, the resulting 30 aligned files were processed by a dedicated measurement software (Hyper Cad S, Cam HyperMill, Open Mind Technologies, Milano, Italy). In this manner, for each analog (n. 120), linear (Δ*X*, Δ*Y* and Δ*Z*-axis) and angular deviations (ΔANGLE) were assessed. The values of linear discrepancies were considered as a single three-dimensional discrepancy value, called the Euclidean distance (ΔEUC).

### 2.5. Statistical Analysis

Continuous variables were summarized by the mean, standard deviation (SD), quantiles and minimum and maximum observed values; categorical variables as absolute and relative frequencies. Multivariable analysis was based on the Analysis of Covariance model. Two different models were fitted, considering Euclidean distance and ANGLE as response variables. In both the models, the main effect of Scan and Position and their interaction were considered. Analysis of covariance tables was reported: main effects and interaction were evaluated using F-statistic. Parameters and standard errors were estimated by the maximum likelihood method; the significance of estimated effects was assessed using t-statistic. All analyses were undertaken using SAS version 9.4 (SAS Institute, Cary, NC, USA) and R version 3.4 (The R Foundation for Statistical Computing, Vienna, Austria).

## 3. Results

The deviations from the reference scan and the 30 test scans were calculated for each analog over the Y, X, Z-axis and angulation (*n* = 120). The alignment of each test scan with the reference scan depends on negative and positive values. By considering the reference axis system used, the data must be interpreted as follows: negative values on the *X*, *Y* and *Z* axis featured a scan body positioned frontward, left and downward, respectively, while the positive ones are in the opposite direction on each axis. The univariate analysis showed the following results: ΔY mean −23 µm (SD 75) ranging from −333 to 65 µm; ΔX average −12 µm (SD 51) ranging from −150 to 83 µm; ΔZ mean −10 µm (SD 39) ranging from −223 to 43 µm. When the linear discrepancy was considered a tridimensional one, ΔEUC reported these results: mean 82 µm (SD 61) ranging from 8 to 347 µm. Lastly, the ΔANGLE error mean was 0.33° (SD 0.20), ranging from 0.02 to 0.92° (Table 1).

The error distribution was centered on the reference measurement for the X, Y and Z-axis, while it was moved towards the right (positive bias) for the angulation and three-dimensional (3D) deviation as they were calculated on absolute values.

For the multivariate analysis, when ΔEUC was considered as a response variable, a significant influence of ISB’s wear by scans in interaction with position 3.6 was identified (*p* < 0.0001), while no significant effect of position and of the scans related to other positions was detected (Table 2) (Figure 5). 

The graph in Figure 6 shows the influence of the three different linear deltas (ΔX-ΔY-ΔZ) of the implant in position 3.6 in correlation to the 30 different scans; the major error is related to ΔY followed by ΔX.

When ΔANGLE was considered as a response variable, a significant influence of ISB’s wear on position 3.6 was identified (*p* < 0.0001), while no significant effect of the other positions and the scans correlated to the position was detected (Table 3) (Figure 7).

## 4. Discussion

Currently, implant-supported prostheses are considered a reliable and effective treatment option to replace natural missing teeth [20]. In order to transfer the three-dimensional implant position, an accurate impression represent a crucial step in implant treatment [21]. IOS was suggested to be a reliable alternative in the clinical practice to conventional impression methods in the manufacture of implant-supported crowns and short-span fixed dental prostheses [22]. Moreover, digital implant impressions could offer advantages over conventional impressions, such as reduced risks of distortion during the impression and laboratory phases, improved patient comfort and acceptance, and improved time efficiency [23,24]. However, digital workflow is still subject to errors, which could derive from the digital impression and CAD/CAM software, as well as production (subtractive or additive) manufacturing processes. These last still depend on the accuracy of the impression and master model. IOS represents a fundamental part of the digital workflow; therefore, accuracy is an essential requirement [6]. The influence of ISB design on the accuracy of IOS is not altogether understood [14]. As mentioned, in single-crown or short-span restorations, digital implant scans using ISBs are shown to have similar accuracy as conventional impressions [25]. However, in completely edentulous patients, a decrease in accuracy was shown [26]. Indeed, in attempting to scan an edentulous arch with ISBs, one specific challenge is the limited number of quality reference points between the scan bodies [27]. ISBs materials and shape were proved to influence the overall accuracy of the digitized data [3,27,28]. To the best of our knowledge, the influence of ISB wear on the accuracy of IOS for the complete arch was never analyzed. Therefore, the purpose of this in vitro trial was precisely to evaluate the influence of PEEK ISBs wear on the overall accuracy of digital impression. 

This was carried out by evaluating the linear and angular deviation of each ISB compared to the reference data that were previously obtained by a reference scan obtained by a dedicated extraoral scanner with high precision. The ISBs were subject to 30 screwing and unscrewing cycles before each corresponding scan, and the deviations were assessed for each ISB for a total of 120 test data. The obtained results showed that the ISB wear, considering ΔEUC, influenced the accuracy of IOS in interaction with the ISB position. In particular, tilted implant analogs’ (3.6–17° and 4.7–30° mesial angulation) accuracy were negatively influenced, with a statistically significant interaction noticed for position 3.6 (*p* < 0.0001). Considering ΔANGLE, a statistically significant effect of position 3.6 was recorded (*p* < 0.0001). These data suggested that ISB base wear could be detrimental to the seating of ISBs on angulated implants. That could be explained by the physical properties of the PEEK material, that as a polymer, could be worn out after repeated mechanical strains, especially at the base level. Furthermore, implants with greater inclination were subjected to loss of accuracy after repeated use, probably because of the higher strain generated during the screwing and unscrewing procedures leading to a collapse of the ISB position towards the inclination direction of the implant. According to the results of the present study, a statistically significant correlation between position and scan number was noticed for position 3.6. For this reason, the distribution of linear discrepancies for implant in the position of 3.6 related to all 30 scans was further investigated, showing a greater deviation contribution on the Y- (lateral) and X-axis (longitudinal), recording the implant position more distally and lingually, precisely towards the implant inclination direction. This would support the hypothesis that an ISB made by a plastic material could be worn out after repeated use and that an ISB with a worn-out base could be detrimental in recording the position of tilted implants, as a structural failure could be recorded according to the direction of the implant tilting. These conclusions are limited to the IOS and ISB material analyzed and do not consider the implications that the sterilization cycles could have on the material wear.

## Figures and Tables

**Figure 1 materials-15-00927-f001:**
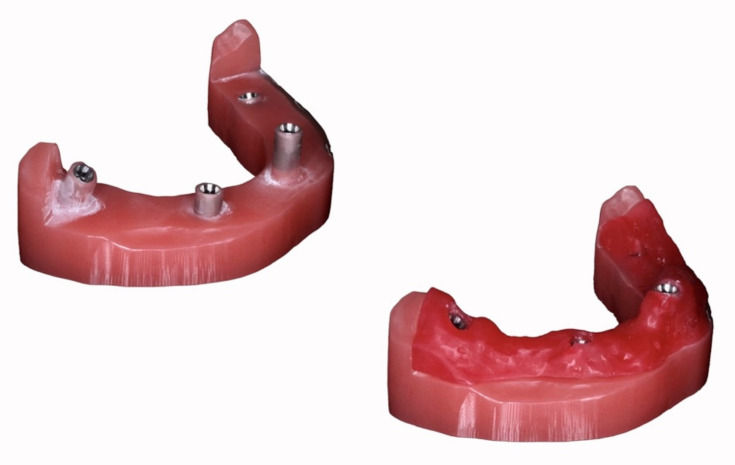
Edentulous mandibular polymethylmethacrylate (PMMA) milled model with and without the removable soft tissue frame.

**Figure 2 materials-15-00927-f002:**
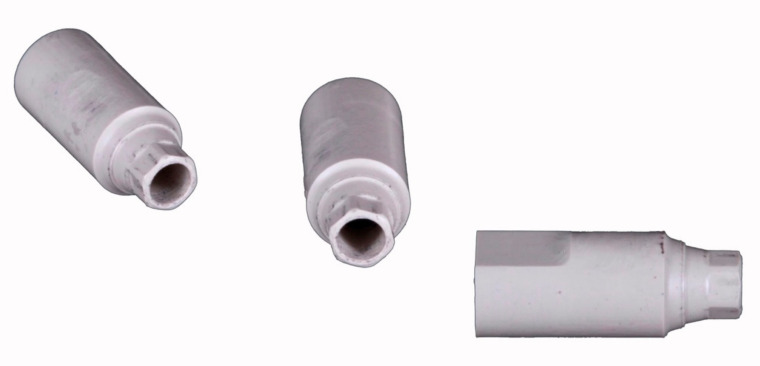
Cylindrical polyetheretherketone (PEEK) implant scanbodies (ISBs).

**Figure 3 materials-15-00927-f003:**
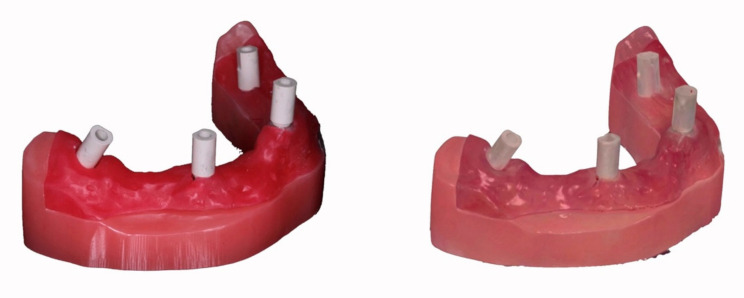
Master model with ISBs on-site and relative IOS scan.

**Figure 4 materials-15-00927-f004:**
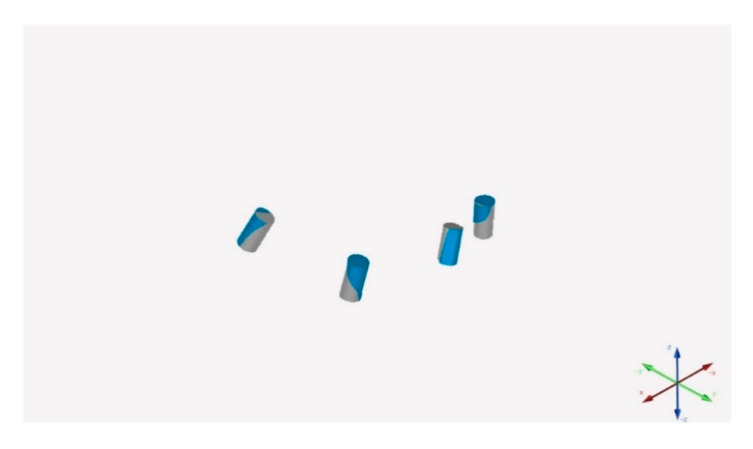
Reference file aligned with test files.

**Figure 5 materials-15-00927-f005:**
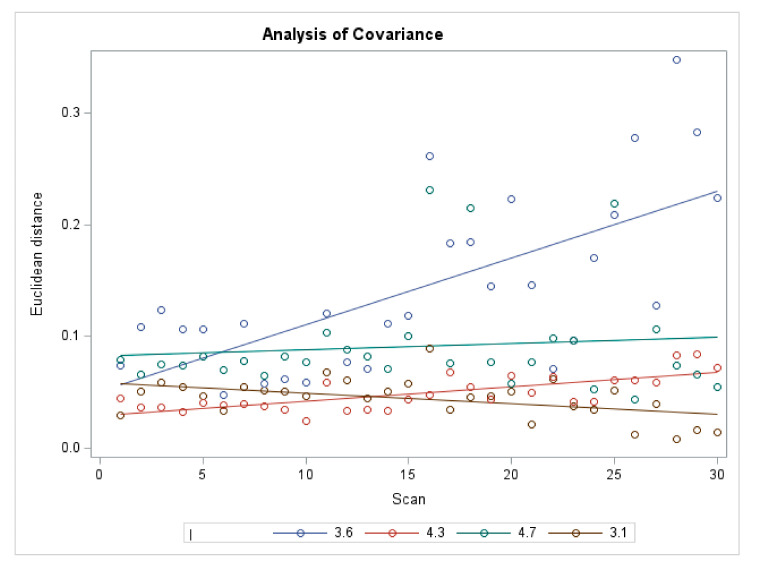
Analysis of Covariance for ΔEUC.

**Figure 6 materials-15-00927-f006:**
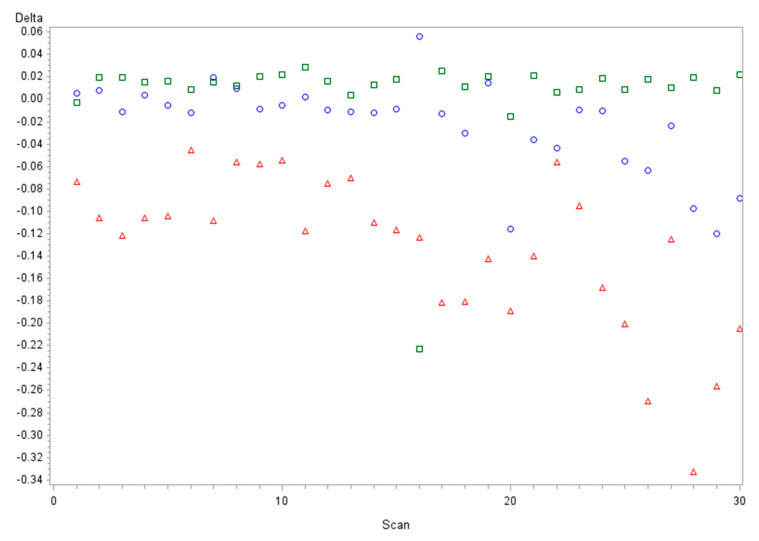
Distribution of linear discrepancies for implant in position of 3.6 related to all 30 scans. (ΔX = circle blue, ΔY = triangle red, ΔZ = square green).

**Figure 7 materials-15-00927-f007:**
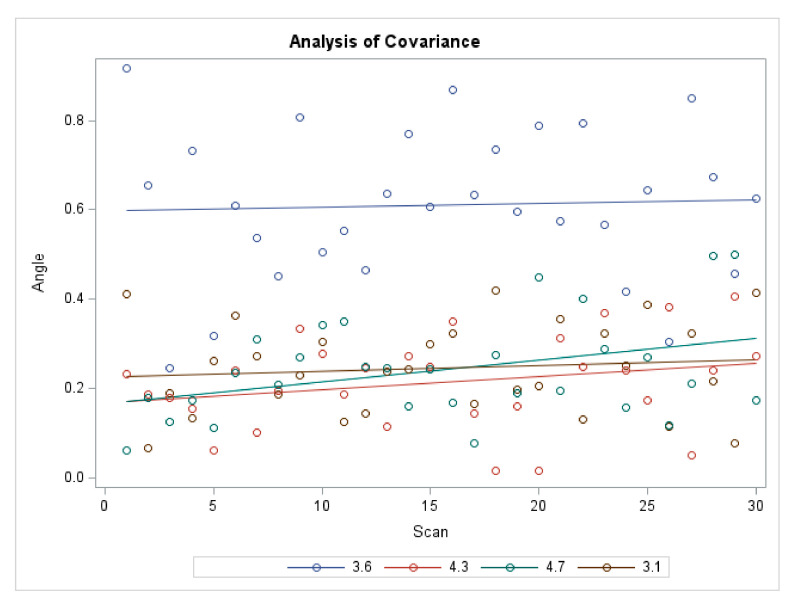
Analysis of Covariance for ΔAngle.

**Table 1 materials-15-00927-t001:** Fitted Normal Distribution for considered variables.

Variable	Number of Observations	Mean	Std Deviation	Max	Min	Q1	Q2 (Median)	Q3
ΔX (μm)	120	−12	51	83	−150	−57	−3	28
ΔY(μm)	120	−23	75	65	−333	−41	4	24
ΔZ (μm)	120	−10	39	43	−223	−23	−3	11
ΔANGLE (°)	120	0.33	0.21	0.92	0.02	0.18	0.27	0.42
ΔEUC (μm)	120	82	61	347	8	45	63	93

**Table 2 materials-15-00927-t002:** Analysis of covariance. Dependent variable Euclidean distance. (* = interaction).

**Source of Variation**	**DF**	**Mean Square (μm)**	**F Statistic**	***p*-Value**
Scan	1	26.5	17.83	<0.0001
Position	3	3.4	2.31	0.0798
Scan * Position	3	19.9	13.39	<0.0001
**Parameter**	**Estimate (μm)**	**Standard error**	**t statistic**	***p*-value**
**Intercept**	58.9	14.4	4.08	<0.0001
**Position** (reference 3.1)				
3.6	−7.8	20.4	−0.38	0.7032
4.3	−29.9	20.4	−1.47	0.1457
4.7	23.3	20.4	1.14	0.2561
**Scan**	−0.9	0.8	−1.17	0.2432
**Scan * Position** (reference 3.1)				
Scan * Position 3.6	6.9	1.2	6.01	<0.0001
Scan * Position 4.3	2.2	1.2	1.95	0.054
Scan * Position 4.7	1.5	1.2	1.34	0.1843

**Table 3 materials-15-00927-t003:** Analysis of covariance. Dependent variable Angle. (* = interaction).

**Source of Variation**	**DF**	**Mean Square (μm)**	**F Statistic**	***p*-Value**
Scan	1	53.5	3.44	0.0664
Position	3	306.7	19.72	<0.0001
Scan*Position	3	7.6	0.49	0.6917
**Parameter**	**Estimate (μm)**	**Standard error**	**t statistic**	***p*-value**
**Intercept**	227.3	46.7	4.87	<0.0001
**Position** (reference 3.1)				
3.6	371.5	66.1	5.62	<0.0001
4.3	−57.8	66.1	−0.87	0.3837
4.7	−60.6	66.1	−0.92	0.3606
**Scan**	1.2	2.6	0.46	0.6456
**Scan * Position** (reference 3.1)				
Scan * Position 3.6	−0.4	3.7	−0.11	0.9143
Scan * Position 4.3	1.7	3.7	0.45	0.6535
Scan * Position 4.7	3.6	3.7	0.98	0.3316

## Data Availability

The data are privately stored by the authors and can be requested to the corresponding author.

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
