# Peer review of "Influence of Implant Scanbody Wear on the Accuracy of Digital Impression for Complete-Arch: A Randomized In Vitro Trial"

_materials, 2022, doi:10.3390/ma15030927_

Round 1

Reviewer 1 Report

The paper topic  ‘Influence of implant scanbody wear on the accuracy of digital  impression for complete-arch: a randomized in vitro trial’ is appropriate to be published in Materials .

The research concerning ISB are not novel, however the authors were focused on the problem of edentulous arch impressions, that have never been investigated before.

The tables and figures are clear and correct.

The results are described clearly and completely and correspond with the research method.

I suggest publication after the authors have considered the following minor remarks:

  1. The references are relevant to the subject of research, however the number of references (22) is not sufficient.
  2. The introduction section is poor in content – it should be expanded and include:

-short description of types of ISB and the inserting procedure – to identify potential factors that may further cause the inaccuracies during impressions

-line 51 – which materials are used to produce ISB

-line 55 – how may times during the procedure ISB is being  repositioned?

  1. Methods

 -  line 99 – is there any scientific explanation for choice the number of 30 (according to screw and unscrew the ISBs?

  1. Discussion

-Is there any potential to combine microscopic or spectroscopic techniques to assess ISB wear?

  1. Conclusions

There should be separate “Conclusions” section after discussion. It should summarize  the clinical significance and the future perspectives for such kind of research.

Author Response

Dear Reviewer,

First of all, I would like to thank you for your revision. I supplied to integrate the section “Introduction” with the requested information and also add some other references. The ISBs were repositioned 30 times during the procedures, but this information was reporting yet in the section “Materials and methods”. This choice was carried out by a preliminary evaluation of the protocol by a statistician. We are evaluating further research protocols to assess the ISB wear through microscopic or spectroscopic techniques, that could also further validate the evidence emerged in the study. At last, the “conclusions” section is not mandatory, and it can be added to the manuscript, according to the guidelines, if the discussion is unusually long or complex, which does not appear to be such in this paper.

Best regards.

Reviewer 2 Report

Thank you for submitting your research on a new topic in digital dentistry.
I have some questions about your research.

M&M part
1. Did you match the direction of the flat surface of the scan region of the scan body?
2. In Figure 1, indicate the implant fixture size and angulation.

Result & Discussion part
1. Additional discussion should be considered about why only Implant 3 & 6 show different result. In my opinion, result of this study shows lack of regularity, so It is doubtful whether it is an error in the scan body production process.
2. Can a statistically significant difference in this study be considered a clinically significant difference?
3. A literature review will be necessary to see if there have been similar results in other papers that have studied the accuracy of traditional pick up impression coping.
4. The first part of the discussion seems to overlap with the contents of the introduction.

Author Response

Dear Reviewer,

Thank you for your interest in evaluating the study. The alignment between the reference files and the test one, how it is described, was performed by a best fit algorithm according with a 0.01mm alignment tolerance and 2 alignment optimizations, accomplished after the file superimposition. If your interest regards the alignment between the scandody acquired by IOS and its CAD file, yes, this superimposition was carried out aligning the flat surfaces, at the edges of which reference points have been placed for alignment. The major error on implant 3.6 does not seem to be related to the ISBs production process (for which a tolerance of ±0.01mm is declared by the same manufacturer) also because the ISBs have been unscrewed and screwed for each of the 30 cycles of scan in random positions. The statistically significant difference in this study could be considered a clinically significant difference, but how is declare the results of this trial are linked to the study in vitro design; so further studies are necessary to assess these results also in vivo where we expect a major error according to the evidence that edentulous patients are considered the most stresses condition for IOS device. To the best of our knowledge tilted implant represents a challenging condition for the traditional pick-up implant impression, but our result are linked to the wear of ISB and no data are available for this conditions regard traditional implant impression. In the first part of the discussion, we would like to summarize the state of problem to connect it to the evidence emerge by this trial.

Best regards.

Reviewer 3 Report

Based on the results of an in-depth evaluation that I have done for a article with the title “Influence of implant scanbody wear on the accuracy of digital impression for complete-arch: a randomized in vitro trial”, I think this article is very well-designed and this manuscript should be accepted for publication in Materials after minor revision.

  1. Author should made figure presentation more scientific in Figure 1, Figure 2, Figure 3, and Figure 4 by removing the background on the image and providing a detail illustration and description in figure.

  1. I would encourage and advise you to adopt the additional references published by, MDPI related to wear as follow.

The Effect of Bottom Profile Dimples on the Femoral Head on Wear in Metal-on-Metal Total Hip Arthroplasty. J. Funct. Biomater. 2021, 12, 38. https://doi.org/10.3390/jfb12020038

Author Response

Dear Reviewer,

I would like to thank you. I provided to add the reference request. Regarding the images, these have no background and the captions briefly describe what is reported in full in the body of the text.

Best regards.